# Single Excited Dual Band Luminescent Hybrid Carbon Dots-Terbium Chelate Nanothermometer

**DOI:** 10.3390/nano11113080

**Published:** 2021-11-15

**Authors:** Rustem R. Zairov, Alexey P. Dovzhenko, Kirill A. Sarkanich, Irek R. Nizameev, Andrey V. Luzhetskiy, Svetlana N. Sudakova, Sergey N. Podyachev, Vladimir A. Burilov, Ivan M. Vatsouro, Alberto Vomiero, Asiya R. Mustafina

**Affiliations:** 1Arbuzov Institute of Organic and Physical Chemistry, FRC Kazan Scientific Center, Russian Academy of Sciences, Arbuzov Str., 8, 420088 Kazan, Russia; sudakova@iopc.ru (S.N.S.); spodyachev@gmail.com (S.N.P.); asiyamust@mail.ru (A.R.M.); 2Department of Physical Chemistry, Kazan (Volga Region) Federal University, Kremlyovskaya Str., 18, 420008 Kazan, Russia; aleksej_dovghenko@mail.ru (A.P.D.); sarkanich98@mail.ru (K.A.S.); ultrav@bk.ru (V.A.B.); 3Department of Nanotechnologies in Electronics, Kazan National Research Technical University Named after A.N. Tupolev-KAI, 10, K. Marx Str., 420111 Kazan, Russia; irek.rash@gmail.com; 4Federal State Autonomous Educational Institution of Higher Education “Gubkin Russian State University of Oil and Gas” (National Research University), Leninsky Prospect, 65, 119991 Moscow, Russia; luzhetskiy@yandex.ru; 5Department of Chemistry, M. V. Lomonosov Moscow State University, Lenin’s Hills 1, 119991 Moscow, Russia; vatsouro@petrol.chem.msu.ru; 6Department of Molecular Sciences and Nanosystems, Ca’ Foscari University Venezia, Via Torino 155, 30172 Venezia-Mestre, Italy; alberto.vomiero@ltu.se; 7Division of Materials Science, Department of Engineering Sciences and Mathematics, Luleå University of Technology, SE-971 87 Luleå, Sweden

**Keywords:** carbon dots, terbium complex, calix[4]arene, molecular thermometer, nanothermometry, hybrid nanoparticles

## Abstract

The report introduces hybrid polyelectrolyte-stabilized colloids combining blue and green-emitting building blocks, which are citrate carbon dots (CDs) and [TbL]^+^ chelate complexes with 1,3-diketonate derivatives of calix[4]arene. The joint incorporation of green and blue-emitting blocks into the polysodium polystyrenesulfonate (PSS) aggregates is carried out through the solvent-exchange synthetic technique. The coordinative binding between Tb^3+^ centers and CD surface groups in initial DMF solutions both facilitates joint incorporation of [TbL]^+^ complexes and the CDs into the PSS-based nanobeads and affects fluorescence properties of [TbL]^+^ complexes and CDs, as well as their ability for temperature sensing. The variation of the synthetic conditions is represented herein as a tool for tuning the fluorescent response of the blue and green-emitting blocks upon heating and cooling. The revealed regularities enable developing either dual-band luminescent colloids for monitoring temperature changes within 25–50 °C through double color emission or transforming the colloids into ratiometric temperature sensors via simple concentration variation of [TbL]^+^ and CDs in the initial DMF solution. Novel hybrid carbon dots-terbium chelate PSS-based nanoplatform opens an avenue for a new generation of sensitive and customizable single excited dual-band nanothermometers.

## 1. Introduction

The wide diversity of organic and inorganic luminophores allows different mechanisms to generate luminescent responses on external stimuli to be applied, which results in a great diversity of sensing systems [1,2,3]. Progress in sensitivity and selectivity of sensing is a real challenge, which requires further evolution of sensing systems. A ratiometric approach based on the generation of two emission bands with different sensitivities on specific external stimuli opens great opportunities in the evolution of sensing systems [4,5]. Combining luminophoric blocks with different natures within one nanobead is a promising route to develop ratiometric sensing systems, which has been well exemplified by incorporating different luminescent blocks into polymeric nanobeads [6,7,8,9], including non-covalently self-aggregated water-soluble polymers [10,11,12]. Polyelectrolytes provide a very promising type of water-soluble polymers due to the multitude of surface-exposed ionic groups providing high affinity to oppositely charged surfaces or counter-ions [13,14,15]. Counter-ion binding is a well-known factor facilitating their intra- and inter-molecular interactions, in turn favoring coil-like conformation of polyelectrolytes followed by their self-assembly into specific clew-like aggregates [16]. However, fundamental backgrounds underlying the use of polyelectrolyte-based aggregates as promising nanobeads for components with opposite charges and different natures are insufficiently recognized.

All chemical and biological processes are greatly influenced by temperature, which makes temperature measurements crucial in a large number of practical applications. Conventional temperature sensors, so-called contact thermometers, cannot remotely report the local temperature conditions. Moreover, the explosive growth of interest in micro and nano dimensional objects imposes increasingly stringent requirements in thermometric studies. Real-time response and high spatial resolution, in addition to the constraint of having nontoxic and biocompatible devices, makes nanothermometry a challenging area of research in which various functional materials are involved [17,18,19]. Luminescence-based temperature sensing is outstanding due to its merits of being noninvasive and accurate [20,21,22]. Thus, the design of novel bicomponent temperature sensors characterized with two well-resolved and intensity comparable luminescence peaks that possess both strong emission and submicron dimensions can be realized through the ratiometric approach.

Among luminescent nanomaterials, photoluminescent carbon dots (CDs) have emerged as important materials in sensing and cell imaging [23,24] since the high surface charge of CDs results from the nature of the hydrophilic exterior layer and is predominantly constituted by chelating anions. This is the reason for its high affinity to counter-ions, including metal ions [25,26]. This is the reason for using CDs as effective optical sensors [27,28,29], although the sensitivity of CDs to any changes in their counter-ion environment restricts their applicability in accurate intracellular thermal sensing [30]. However, the counter-ion binding of CDs opens a window for further evolution of their sensing ability through both incorporation of CDs into self-aggregated polyelectrolytes and their electrostatic and coordinative binding with metal ions.

Ln^3+^-based species possessing narrow emission bands, large Stokes shift, and high emission quantum yields inspired many reports on Ln^3+^-based thermometers with outstanding sensitivity, temperature resolution, and robustness in the last decade [31,32]. In this regard, the implementation of Ln^3+^-based materials to hybrid nanostructures is a budding strategy for the design of novel ratiometric nanothermometers. The difference in the main mechanisms responsible for luminescent response generation by lanthanide chelates and CDs, as well as the possibility to link these components through electrostatic and coordinative binding, provides a prerequisite for their incorporation into polyelectrolyte-based nanobeads as a route for developing ratiometric sensing systems.

Herein, we report a facile strategy for the synthesis of novel hybrid polystyrenesulfonate(PSS)-coated nanoparticles consisting of carbon dots (CDs) and Tb^3+^ complex as a single excited dual-band luminescent nanothermometer. The ligand environment of the Tb^3+^ complex should fit specific requirements, such as (1) high complex stability, (2) incomplete coordination sphere of Tb^3+^ ions, (3) efficient and temperature-responsive Tb^3+^-centered luminescence. The [TbL]^+^ complex with 1,3-diketonate derivatives of calix[4]arene (L) fits well to the aforesaid requirements. The capacity of the hybrid nanoparticles to simultaneously emit in the blue and green regions of the spectrum originates from two different emissive components: CDs and [TbL]^+^ complex, correspondingly. The coordinative binding of green-emitting [TbL]^+^ complex with blue-emitting CDs is highlighted as a tool for their combination within the PSS-nanoplatform, as is schematically demonstrated in Figure 1.

The variation of the concentration conditions will be introduced as a tool to control the mutual influence of the components on their sensing properties in order to develop colloids for temperature monitoring in the 25–50 °C range through the differently emitting registration channels and to transform the colloids into ratiometric temperature sensors.

## 2. Experimental Section

Materials: citric acid (99%), ethylenediamine (99%), Tb(NO_3_)_3_·6H_2_O (99.9 wt%), Triethylamine (TEA) (99%), *N*,*N*-dimethylformamide (DMF), Poly(sodium 4-styrenesulfonate) (PSS) (MWaverage = 70,000), and sodium chloride (NaCl) were purchased from Acros Organics. TRIS-buffer, acetic acid, sodium hydroxide, bovine serum albumin (BSA), DMF were twice distilled over P_2_O_5_. Other reagents were used without additional purification.

Tetra-acetylacetonyl derivative of calix[4]arene with hydroxyl substituents (L, 5,11,17,23-tetrakis[(acetylaceton-3-yl)methyl)]-25,26,27,28-tetrahydroxy-calix[4]arene, Figure 4a) at its lower rims was synthesized in accordance with the previously reported procedure [33].

Synthesis of CDs. Carbon quantum dots were synthesized by microwave synthesis. In a 50 mL flask, 1 g of citric acid was dissolved in 10 mL of distilled H_2_O, and 0.348 mL of ethylenediamine was added. The resulting solution was incubated for 1 min 30 s in a domestic microwave oven until a red caramel residue formed. The product was dissolved in 20 mL of distilled H_2_O, filtered from large particles through a membrane (0.22 μm), and dried at room temperature.

Synthesis of polyelectrolyte nanoparticles PSS-[TbL]. The [TbL]^+^ complex was obtained via mixing Tb(NO_3_)_3_·6H_2_O (C = 0.5 mM) with the ligand in a 1:1 ratio and adding triethylamine (C = 3.0 mM) to deprotonate the β-diketonate groups of the calix[4]arene ligand L. PSS-[TbL] nanoparticles were synthesized by adsorption of PSS polyelectrolyte on the surface nanosized templates, obtained by precipitation of the [TbL]^+^ complex from a DMF solution into an aqueous PSS solution (C = 1 g/L, V = 2.5 mL) and NaCl (C = 0.5 M). To an aqueous solution of PSS was added 0.5 mL of a complex-containing solution of DMF with vigorous stirring (2200 rpm). The resulting colloidal solution was sonicated for 30 min while maintaining a constant temperature of 25 °C. Excess PSS was separated by centrifugation at 4 °C and 15,000 rpm for 15 min, followed by draining the supernatant.

Synthesis of hybrid nanoparticles PSS-{CDs-[TbL]}. Hybrid PSS-{CDs-[TbL]} nanoparticles were synthesized on the basis of a DMF solution containing a TbL and CD complex. In the *Synthesis_1* concentration of [TbL]^+^—4.5 mM and CD—4.5 g/L were used. The concentrations of the components in the *Synthesis_2* were 4.5 mM and 1.5 g/L for [TbL]^+^ and CD, respectively. Corresponding DMF solution ([TbL]^+^ and CD mixture) in an amount of 0.5 mL was added to an aqueous solution of PSS (C = 1 g/L, V = 2.5 mL) and NaCl (C = 0.5 M) with intensive stirring (2200 rpm). PSS was wound onto the resulting nanoscale [TbL]^+^ –CD precipitate with the formation of a stable colloidal solution. Purification of nanoparticles from excess PSS was carried out by centrifugation under conditions identical to those in the synthesis of PSS-[TbL].

The pH variations at 4.5–7.0 and 7.0–8.4 were performed using acetic-acetate and TRIS-buffers, correspondingly.

The UV–Vis measurements were performed on a Lambda 35 spectrophotometer (PerkinElmer, Waltham, MA, USA) using 10 mm cuvettes at room temperature. The steady-state emission and excitation spectra were recorded on a fluorescence spectrophotometer Hitachi F-7100 (Tokyo, Japan) with stigmatic concave diffraction grating. Excitation of samples was performed at 330 nm, and emission detected at 545 nm with 5 nm slit width for both excitation and emission. All samples were ultrasonicated for 15 min before measurements. The dynamic light scattering (DLS) measurements were carried out by means of the Malvern Mastersize 2000 particle analyzer. A He–Ne laser operating at 633 nm wavelength and emitting vertically polarized light was used as a light source. The measured autocorrelation functions were analyzed by Malvern DTS software and the second-order cumulant expansion methods. The effective hydrodynamic diameter (d) was calculated by the Einstein–Stokes relation from the first cumulant: D = kBT/3πηd, where D is the diffusion coefficient, kB is the Boltzmann constant, T is the absolute temperature, and η is the viscosity. The Nano-ZS (MALVERN) with laser Doppler velocimetry and phase analysis light scattering was used to determine the zeta potential. The zeta potential was measured at least three times for each sample. Only multiply reproducible results were taken into account; thereby, they differed by less than 4%. The transmission electron microscopy (TEM) images were obtained with Hitachi HT7700, Japan. The images were acquired at an accelerating voltage of 100 kV. Samples were sonicated in water for 30 min and then dispersed on 200 mesh copper grids with continuous formvar support films.

## 3. Results and Discussion

The synthesis of hybrid carbon dots-terbium chelate nanothermometer owning blue and green-emitting components into the PSS-based nanoplatform was performed through several steps: (1) synthesis of the components; (2) evaluation of their sensing ability to heating and cooling; (3) revealing of the binding between the components; (4) optimization of conditions for the combined incorporation of both components into the PSS-nanoplatform.

### 3.1. Synthesis, Characterization, and Sensing Properties of CDs

Carbon dots (CDs) were synthesized according to a previously reported method in a one-step microwave-assisted reaction of citric acid and ethylenediamine [34]. Transmission electron microscopy (TEM) particle characterization showed that the dots are quasi-spherical monodisperse species with an average size of (2.3 ± 0.6) nm remaining erratically distributed on the substrate surface (Figure 2a). The CDs showed no tendency to agglomerate upon drying on the formvar coated copper grid used as a substrate.

Dispersion in water at a concentration of 10 μg mL^−1^ resulted in a transparent yellowish aqueous colloid stable for at least three days (Appendix A). UV-Vis absorbance spectroscopy revealed that the water solution of CDs has two distinct absorption bands at 350 nm and 270 nm (Figure 2b) assigned to n→π* and π→π* transitions of the aromatic sp^2^ domains, respectively.

Aqueous colloids of CDs exhibit bright blue fluorescence with the maximum at 450 nm as a result of 360 nm xenon lamp excitation (Figure 2b). The color coordinates for luminescence of CDs in H_2_O are shown on the chromaticity diagram (Figure 3a). The CD emission spectrum was found to be dependent of temperature changes in the range of 25–70 Celsius degrees (Figure 3b). Temperature-fluorescence intensity shows linear response (R^2^ = 0.99577 for heating; 0.99548 for cooling) over the entire analysis range, and the temperature sensitivity was determined to be as high as 1.25% °C^−1^. This value is comparable with the majority of other CD-based nanothermometers [35,36,37,38,39].

Temperature-dependent fluorescence of the CDs aqueous colloids is fully reversible within the 25–70 °C temperature range (Figure 3c). The stability of the CDs colloids under the thermal cycling experiments was evaluated by measuring fluorescence intensity recovery through multiple heating and cooling cycles from 35 °C to 45 °C (Figure 3d). The narrowing of the temperature range to 35–45 °C was aimed at physiologically relevant conditions. The thermal reversibility of the CDs was revealed for at least 6 heating-cooling cycles indicating their ability to serve as a fluorescence-based nanothermometer. However, one of the issues of single-parameter thermometers is the need for external calibration to obtain an absolute value of the temperature through the intensity measurement.

### 3.2. Synthesis, Characterization, and Sensing Properties of PSS-[TbL]

As previously reported, the synthesis of hydrophilic colloids from the terbium complexes with cyclophanic 1,3-diketonates is based on their efficient self-assembly in the synthetic conditions of the solvent-exchange technique [5,6,7]. The [TbL]^+^ complex derives from efficient bis-chelating of Tb^3+^ via two diketonate groups of the tetra-1,3-diketone calix[4]arene (Figure 4a,b). The 1:1 complex is formed in the DMF solutions basified by triethylamine (TEA), which is followed by the sensitization of the Tb^3+^-centered luminescence due to the efficient ligand-to-metal energy transfer. Both pronounced and reversible temperature-induced changes of the luminescence of the complex in the DMF solutions (Appendix A) are the reason for its choice as the challenging building block of the PSS-stabilized colloids.

The PSS-[TbL] colloids were synthesized through the solvent exchange procedure [40] (for more details, see the Exp. Section). The synthesized colloids were characterized by means of DLS data (Table 1) and TEM images (Figure 4b). The average size revealed from the DLS measurements is about 200 nm with a rather low polydispersity (PDI is below 0.2). The negative surface charge manifested by the electrokinetic potential value at −20 mV confirms that [TbL]^+^-based colloids are embedded into the PSS-aggregates. The difference between the average sizes revealed by TEM and DLS is in agreement with the above-mentioned assumption that PSS-based aggregates incorporate [TbL]^+^-based colloids.

The steady-state luminescence of PSS-[TbL] nanoparticles is manifested by four narrow bands ^5^D_4_→^7^F_n_ peculiar for Tb^3+^-centered luminescence giving bright green emission with the color coordinates exemplified in Figure 5a. The temperature-dependent luminescence of polystyrenesulfonate-coated PSS-[TbL] colloids was studied for the first time here. Gradual decrease of all four Tb bands is observed upon heating from 25 °C to 70 °C (Figure 5b). The band at 547 nm (^5^D_4_→^7^F_5_) undergoes the highest spectral changes with the sensitivity S_j_ = 3.55% °C^−1^. Cooling the sample back to 25 °C results in luminescence intensity restoration to initial values. The temperature-dependent luminescence is linear (R^2^ = 0.97593 for heating; 0.98803 for cooling) and fully reversible, as can be seen from Figure 5c, where the luminescence under heating or cooling is normalized to the maximum intensity measured at 25 °C. Similar to CDs, luminescence intensity changes of 547 nm band were recorded for PSS-[TbL] within the 35–45 °C range. PSS-[TbL] water colloids demonstrate reversibility for at least six cycles (Figure 5d).

Relative thermal sensitivities for [TbL]^+^ in DMF, CDs, and PSS-[TbL], in aqueous solutions were plotted in Figure 6a. The comparison indicates that the thermal sensitivity of [TbL]^+^ up to S_j_ = 7.68% °C^−1^ at 50 °C exhibits two-fold lowering to S_j_ = 3.55% °C^−1^ under the transformation of [TbL]^+^ to PSS-[TbL] aqueous colloids. Substantial deviation in the thermal sensitivities derives from the difference between the complexes in the DMF solutions and those in the aqueous colloids.

Insignificant luminescence intensity dependence of the main 545 nm band reveals high kinetic stability of PSS-[TbL] in H_2_O to pH changes in the physiological range (pH = 4.5–8.0) (Figure 6b). Full reversibility of I_lum_ upon basification with NaOH and acidification with HCl evidenced no [TbL]^+^ degradation, which makes these colloids a good basis for the monitoring of temperature changes in biological liquids or tissues known to have different acidity. Luminescence λ_em_ = 450 nm band of CDs was also found to be non-sensitive to pH changes at pH = 4.5–8.0 (Appendix A).

### 3.3. Interactions between CDs and [TbL]^+^ in the DMF Solutions

It is worth noting that the incorporation of the positively charged complexes into the PSS-platform is electrostatically facilitated, while the negative charge of CDs is the factor raising the repulsive interactions with the negatively charged nanobeads. However, the counter-ion binding of the CDs in the basified solutions of [TbL]^+^ can facilitate their joint incorporation into the PSS-nanobeads. Thus, interactions between [TbL]^+^ complexes and nanosized carbon dots in the basified DMF solutions should be revealed as the prerequisite of their joint nanoprecipitation in the aqueous-DMF solutions of PSS. The mixing of the components in the basified DMF solutions results in the dual blue-green fluorescence under the excitation by 330 nm (Figure 7a,b). The spectra represented in Figure 7b demonstrate a gradual decrease of Tb-centered luminescence intensity upon the addition of CDs, which amplifies their interaction with [TbL]^+^ complex. Thus, careful analysis of the fluorescence of each component under concentration variation of another one has been undertaken and demonstrated in Figure 7c,d.

As it was previously reported, the basification is aimed at shifting the complex formation equilibrium towards [TbL]^+^ [12,33], where Tb^3+^ ions are bis-chelated with two 1,3-diketonate moieties. Basified conditions facilitate deprotonation of the surface groups of CDs, which, in turn, is followed by their counter-ion binding with both triethylammonium and [TbL]^+^ cations. The fluorescence of CDs has been studied in the DMF solutions at various concentrations of TEA and Tb^3+^ ions to recognize and separate their effects. Basification of CDs DMF solution by 80 μM of TEA results in a 10% loss of CDs fluorescence intensity (Figure 7c). This fact points to high deprotonation extent of various groups localized on the surface of carbon dots. The addition of Tb^3+^ ions to TEA rich conditions (80 μM) reveals twofold fluorescence intensity quenching of CDs 450 nm peak, indicating efficient binding with the surface of CDs, which prevents Tb^3+^ ions from instantaneous hydrolysis (Figure 7c).

The efficient quenching of the CD-centered fluorescence by the Tb^3+^ ions indicates their efficient complex formation with the chelating anions localized at the surface of CDs. Thus, the complex formation should trigger the stripping of the Tb^3+^ ions from [TbL]^+^ complex, which can be introduced schematically by the *Equilibrium 1*. However, both the complexing ability of the CDs and incomplete saturation of the inner-coordination sphere of [TbL]^+^ complex allow assuming the probability of the ternary complex formation due to *Equilibrium 2*. Both *Equilibriums* are designated in schemes illustrated in Figure 7e,f. The observed changes in the Tb-centered luminescence derive from the interference of *equilibriums 1* and *2* since the luminescence quenching due to equilibrium (1) can be in some extent compensated by the ternary complex formation in accordance with equilibrium (2), since the excited levels of the CDs lie above the excited Tb^3+^-centered level as it is schematically shown in Appendix A.
[TbL] + CD → L + CDTb(1)
[TbL] + CD → [TbLCD](2)

Monitoring of the steady-state and time-resolved luminescence of the complex under the varied concentration of the CDs in the DMF solutions can help in choosing optimal concentration to minimize the undesirable destruction of [TbL]^+^ complex and to adjust the intensities of the CDs- and Tb^3+^-centered luminescence on the comparable levels under the same excitation wavelength (330 nm). The observed losses in the 547 nm peak intensity of Tb-centered luminescence (Figure 7d) under the increased concentration of the added CDs point to partial degradation of [TbL]^+^ complexes. The excited-state lifetime values evaluated from the time-resolved luminescence measurements reveal their insignificant changes under the growing concentrations of CDs at constant C_[TbL]_ = 0.45 mM (Appendix A). This confirms the partial stripping of Tb^3+^ ions from [TbL]^+^ complexes by the CDs as the only reason for the quenching of the luminescence. The quenching of the CD-centered fluorescence under the increased concentration of the added [TbL]^+^ complex also argues for the coordination of either Tb^3+^ ions or [TbL]^+^ complex with the chelating anions of the CDs. The binding with Tb^3+^ and [TbL]^+^ as the counter-ions should induce a surface charge neutralization of the CDs, in turn, facilitating their joint incorporation with [TbL]^+^ complex into the PSS-based aggregates.

### 3.4. Synthesis and Characterization of PSS-{CDs-[TbL]} Colloids

Synthesis of the hybrid PSS-{CDs-[TbL]} colloids was performed through the solvent exchange method (see Experimental Section for more details). Variation of the component’s concentration, CDs and [TbL]^+^, in initial DMF solution enables to vary the intensity ratios of 450 nm and 547 nm emission peaks upon single excitation by 360 nm wavelength, correspondingly (Figure 8). It is worth noting that the easy phase separation of the PSS-stabilized colloids through centrifugation facilitates washing out the undesired admixtures and residual unbound CDs from PSS-{CDs-[TbL]} colloids. Two examples of PSS-{CDs-[TbL]} hybrids have been synthesized at various [TbL]^+^ and CDs concentrations. The synthetic conditions 1.5 mM [TbL]^+^ and 1.5 g∙L^−1^ CDs for TbL and CD, as well as 4.5 mM [TbL]^+^ and 1.5 g∙L^−1^ CDs will be herein and further designated as *Synthesis_1* and *Synthesis_2*. It is worth noting the phase separation through centrifugation as the key step of the synthesis of PSS-{CDs-[TbL]} colloids since the colloids of CDs can’t be separated from the aqueous solutions. Thus, the appearance of the dual blue-green emitting components in the redispersed after the phase separation aqueous colloids provides a clear indication of the joint incorporation of both components into PSS-{CDs-[TbL]} colloids (Figure 8). The analysis of the dual luminescence of PSS-{CDs-[TbL]} indicates that the I_547_/I_450_ bands ratio can be varied in a wide range from 1.34 to 8.02 enabling smooth tuning of green and blue contribution. It is worth noting that the concentration variation allows changing the steady-state Tb^3+^-luminescence intensity with remaining the excited state lifetime values unchanged (Appendix A), which supports a similarity in the interactions between CDs and [TbL]^+^ within PSS-{CDs-[TbL]} colloids.

The TEM image of PSS-{CDs-[TbL]} (*Synthesis_1*) illustrates the presence of worm-like structures formed from globular species intertwined together and layered onto each other upon drying (Figure 8c). The more layers, the darker the grey color of worm-like structures formed onto the light grey plain color of the formvar grid. Figure 8d represents an enlarged area of the black rectangle from Figure 8c selected for detailed analysis. Black 2–3 nm dots scattered within the structural element of colloidal particles visualize the presence of CDs in PSS-{CDs-[TbL]} colloids proving the hybrid nature of the latter.

Hydrodynamic diameters of PSS-{CDs-[TbL]} hybrids are higher compared to single PSS-[TbL] colloids: (285.2 ± 1.2) nm versus (193.3 ± 1.2) nm; however, PDI values of the hybrid and initial colloids are similar (Table 1). This, along with the higher negative value of zeta potential of the hybrid colloids versus that of the initial colloidal systems (Table 1), provides one more argument for the coexistence of CDs and [TbL]^+^ blocks within each nanoarchitecture. The invariance in luminescence intensity for up to 3 days of storage evidences steadiness of water colloids (Appendix A).

### 3.5. Sensing Properties of PSS-{CDs-[TbL]}

As it was described in the previous paragraphs, both CDs and [TbL]^+^ can give explicit luminescent responses to temperature variation. Fabrication of novel composite nanomaterial out of two temperature-sensitive building blocks embodies the dual-band nanothermometer. Different relative sensitivities of blue (450 nm) and green (547 nm) emission bands and variable ratio between the building blocks make possible the color tuning of PSS-{CDs-[TbL]} depending on the goal of study and complexity of an investigated system. Moreover, the coordinative binding between the CDs and the complexes is the reason for modifying both blue and green luminescent responses to temperature changes. In particular, the heating of PSS-{CDs-[TbL]} (*Synthesis_1*) up to 70 °C results in the irreversible quenching of the Tb^3+^-luminescence, while the temperature responsivity of the CDs components is slightly decreased, but still very strong (Figure 9a). The temperature dependence of the dual emission is illustrated by the chromaticity diagram in Appendix A. Noteworthy, the green luminescence response of PSS-{CDs-[TbL]} (*Synthesis_1*) is reversible within 34–45 °C and can be recycled at least 6 times at least (Figure 9b). This means that two registration channels exist for the temperature sensing with the use of PSS-{CDs-[TbL]} (*Synthesis_1*), which can be selected by the researcher in accordance with an object of research.

The use of PSS-{CDs-[TbL]} (*Synthesis_2*) enables the transformation of the colloids to ratiometric nanothermometers for the temperature sensing applications related to inhomogeneous systems, which are the majority of biological samples (blood, cells, etc.). Minimizing the content of the CDs on going from the colloids (*Synthesis_1*) to those (*Synthesis_2*) enables to suppress dramatically the fluorescent response of the CDs to temperature changing (Figure 9e). The binding of the CDs with Tb^3+^ ions and [TbL]^+^ complexes to a greater extent is facilitated by the conditions of *Synthesis_2* vs. *Synthesis_1.* In turn, the binding event is the reason for the disturbance of both fluorescence of the CDs (Figure 7d) and the responsivity of the fluorescence on the temperature changes (Figure 9c). The green 547 nm band in PSS-{CDs-[TbL]} colloids (*Synthesis_2*) remains highly sensitive to the changes of temperature in the range of 25–50 °C, while the fluorescence of the CDs exhibits the insignificant changes within the temperature range in comparison with the intensity changes of the green band at 547 nm (Figure 9e). This enables to correlate the intensities ratio at 547 and 450 nm with the temperature values in the range of 30–50 °C, which is demonstrated in Figure 9d. The relative sensitivities S_j_ = 2.89%C^−1^ are close to each other for the colloids synthesized in both synthetic conditions. Four heating-cooling cycles performed at alternating temperatures of 30 °C and 50 °C indicate full reversibility of the luminescent response (Figure 9f). It is worth noting the previously reported invariance of the luminescence signal of the PSS-stabilized colloids based on the terbium complexes with calix[4]arene 1,3-diketones in the solutions modeling blood serum [41]. Both low cytotoxicity and convenient cell internalization of these colloids [10] provide a background of applicability of PSS-{CDs-[TbL]} colloids in the monitoring of intracellular temperature. Moreover, the joint incorporation of CDs and [TbL] into the PSS-nanobeads is a prerequisite for their joint localization in one cellular compartment, which, in turn, allows to monitor temperature changes within this cellular compartment through the dual-band emission, where thermally non-sensitive (weak-sensitive) band is referred to local concentration of the particles and second band is responsible for thermosensing. However, additional studies, including cell internalization study, intracellular optical thermometry, cell viability tests, will be performed in the near future and reported elsewhere to outline the applicability of the produced hybrid colloids for the intracellular sensing and/or marking.

## 4. Conclusions

The results introduce joint incorporation of green and blue emitting blocks of different nature exemplified by [TbL]^+^ complexes and CDs into the polysodium polystyrolesulfonate (PSS) aggregates through the solvent-exchange synthetic technique. The coordinative binding between Tb^3+^ centers and the surface groups of CDs is the main driving force of their joint incorporation into the PSS-based nanobeads. The binding between the green and blue-emitting components affects their fluorescence intensity, as well as the ability to give the fluorescent response to the temperature changes. The revealed regularities both enable the development of the colloids for the monitoring of temperature change within 25–50 °C through blue and green-emitting registration channels and to transform the colloids into ratiometric temperature sensors through the easy variation of the synthetic conditions. Novel hybrid carbon dots-terbium chelate-based platform exhibiting single excited dual-band emission opens an avenue for a new generation of sensitive and customizable nanothermometers.

## Figures and Tables

**Figure 1 nanomaterials-11-03080-f001:**
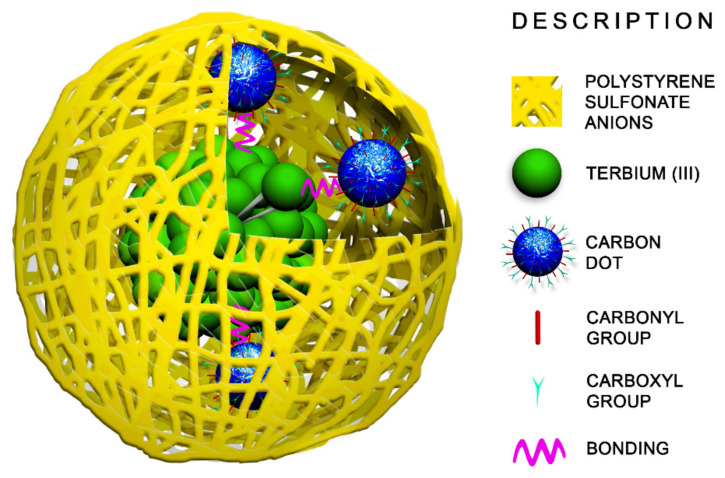
Schematic illustration of PSS-{CDs-[TbL]} hybrids.

**Figure 2 nanomaterials-11-03080-f002:**
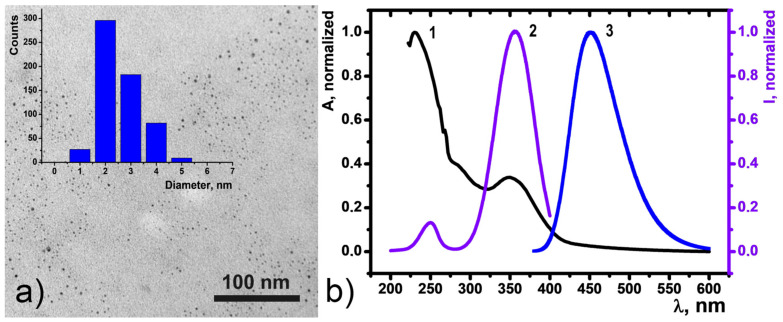
(**a**) TEM image of dried CDs and corresponding size distribution diagram (inset). (**b**) UV-Vis absorption spectrum (1), room temperature excitation (2, λ_em_ = 450 nm) and fluorescence (3, λ_ex_ = 360 nm) spectra of a 10 mg mL^−1^ CD aqueous dispersion.

**Figure 3 nanomaterials-11-03080-f003:**
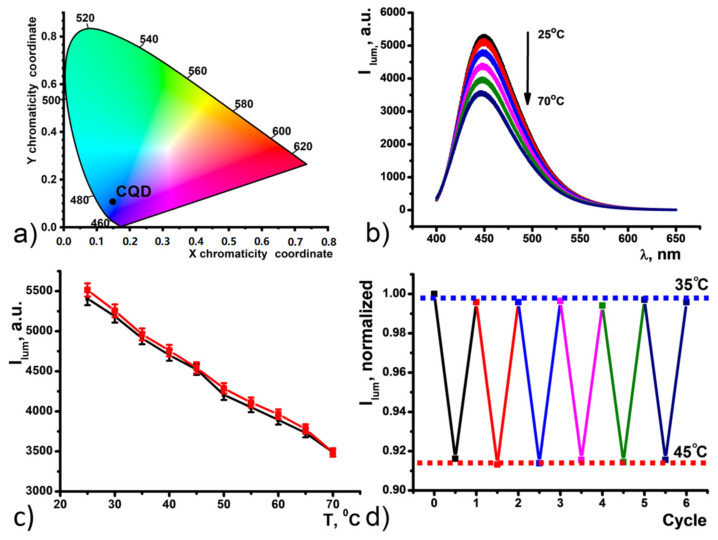
(**a**) Chromaticity diagram for CDs in H_2_O. (**b**) Temperature dependence of CDs emission upon heating. (**c**) Changes in the fluorescence intensity of the CDs (λ_ex_ = 360 nm) as a function of temperature under the heating (black) and cooling (red) over the 25–70 °C range. (**d**) Normalized fluorescence intensity upon 6 heating-cooling cycles of CDs under an alternating temperature of 35 °C and 45 °C.

**Figure 4 nanomaterials-11-03080-f004:**
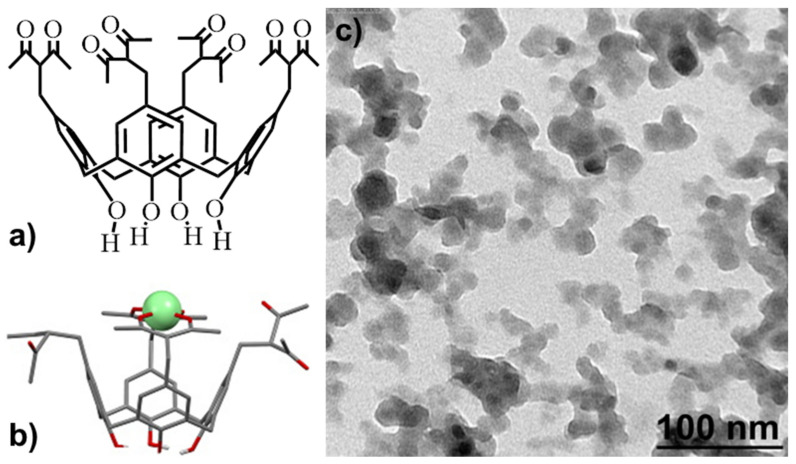
Structure of (**a**) L and (**b**) [TbL]^+^. Reprinted with permission from Ref. [33]. Copyright 2016 Elsevier. (**c**) TEM image of dried PSS-[TbL] nanoparticles.

**Figure 5 nanomaterials-11-03080-f005:**
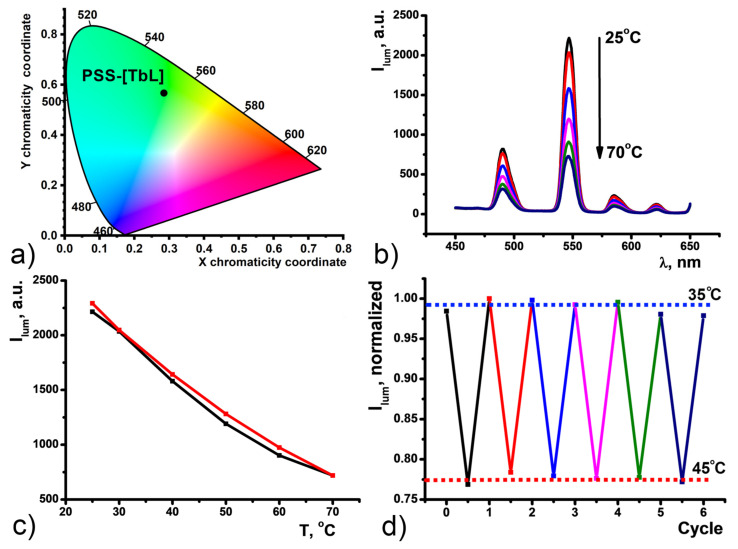
(**a**) Chromaticity diagram for PSS-[TbL]. (**b**) Temperature dependence of PSS-[TbL] emission spectrum upon heating. (**c**) Changes in the fluorescence intensity of the PSS-[TbL] (l_ex_ = 360 nm) as a function of temperature under the heating (red) and cooling (black) within the 25–70 °C range. (**d**) Normalized fluorescence intensity upon six heating-cooling cycles of PSS-[TbL] under alternating temperatures of 35 °C and 45 °C.

**Figure 6 nanomaterials-11-03080-f006:**
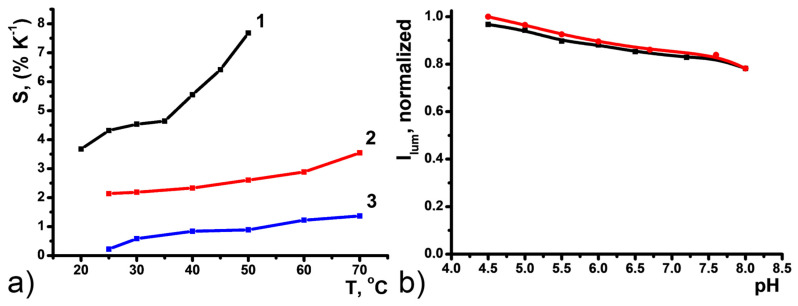
(**a**) Relative thermal sensitivities (multiplied by −1) for [TbL]^+^ in DMF (1), PSS-[TbL] in H_2_O (2), and CDs in H_2_O (3). (**b**) Normalized luminescence of PSS-[TbL] versus pH.

**Figure 7 nanomaterials-11-03080-f007:**
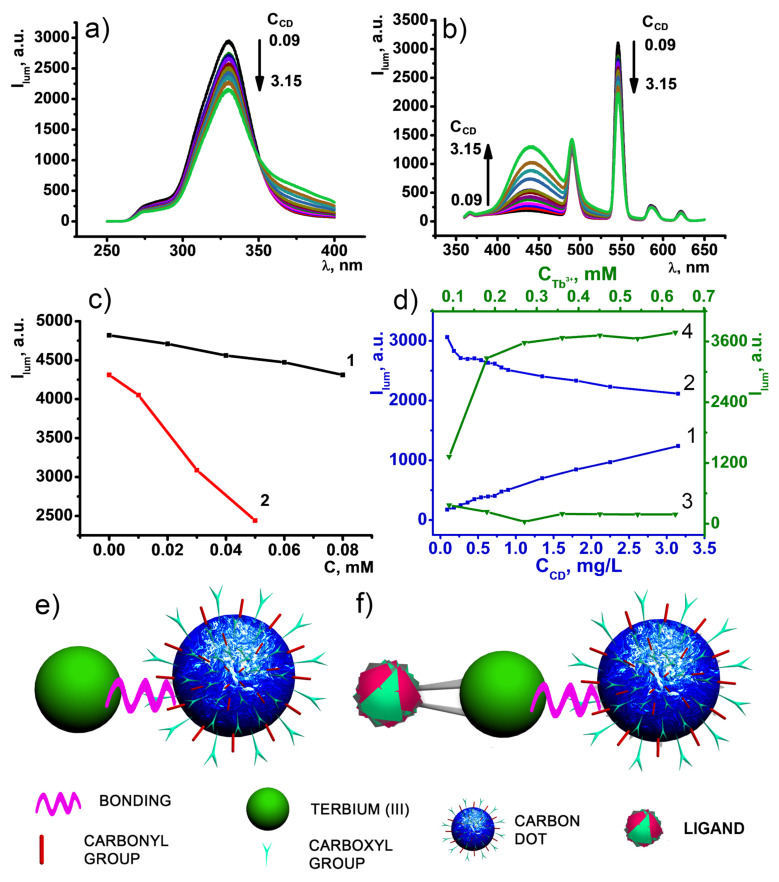
(**a**) Excitation (λ_em_ = 547 nm) and (**b**) emission (λ_ex_ = 330 nm) spectra of CDs and [TbL]^+^ in DMF at different CDs to [TbL]^+^ ratio (C_CD_ = 0.09–3.15 mg⋅L^−1^; C_[TbL]_ = 0.45 mM). (**c**) Fluorescence intensity of CDs (0.01 g/L) versus C_TEA_ (1) and versus C_Tb_ (2) in the presence of 0.08 mM TEA in DMF. (**d**) Intensity of 450 nm (1) and 547 nm (2) emission peaks as a function of C_CDs_ at constant C_[TbL]_ = 0.45 mM. Intensity of 450 nm (3) and 547 nm (4) emission peaks as a function of C_[TbL]+_ at constant C_CD_ = 4.5 mg⋅L^−1^ in DMF. (**e**) Schematic view of Tb-L interaction exemplified by *Equilibrium 1*. (**f**) Schematic view of CD-Tb-L interaction exemplified by *Equilibrium 2*.

**Figure 8 nanomaterials-11-03080-f008:**
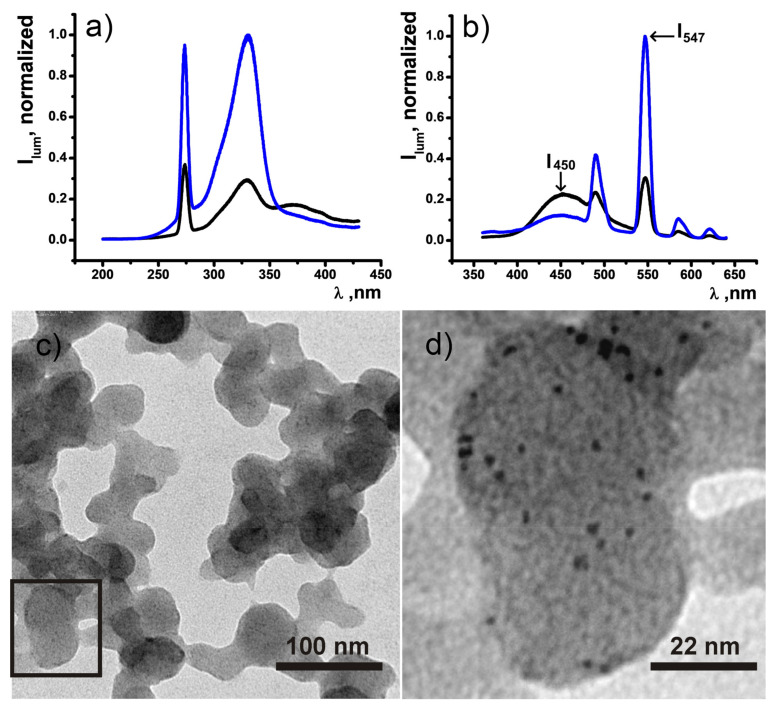
(**a**) Excitation spectra of PSS-{CDs-[TbL]} in H_2_O, (λ_em_ = 450 nm and 547 nm) (**b**) Emission spectra of PSS-{CDs-[TbL]} synthesized according to the *Synthesis_1* and *Synthesis_2* (λ_ex_ = 360 nm). (**c**) TEM image of dried PSS-{CDs-[TbL]} colloids. (**d**) Enlarged black rectangle area from the Figure 8c.

**Figure 9 nanomaterials-11-03080-f009:**
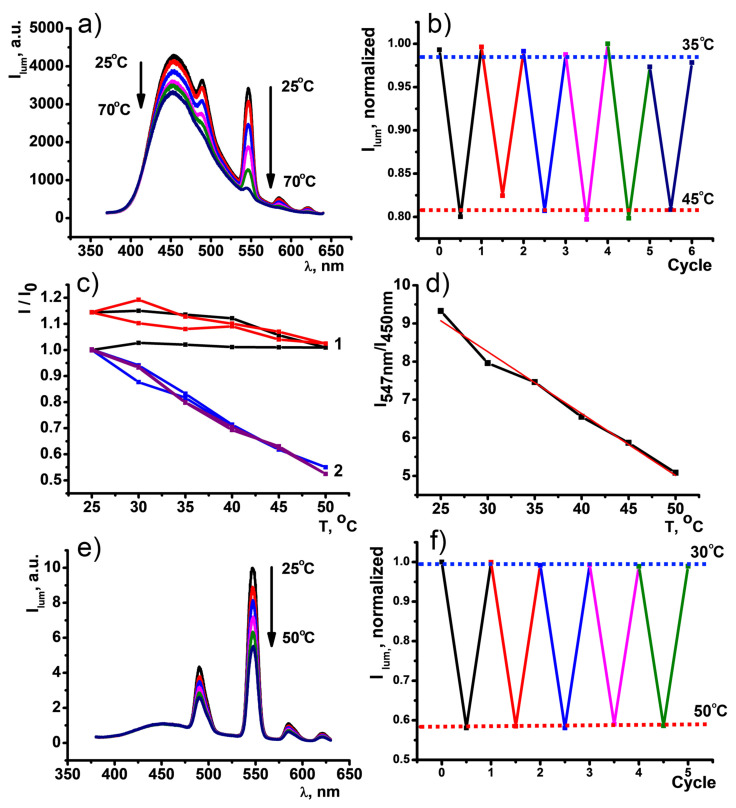
(**a**) Emission spectra of the PSS-{CDs-[TbL]} (*Synthesis_1*) (l_ex_ = 360 nm) as a function of temperature over the 25–50 °C range. (**b**) Temperature dependence of 547 nm band of PSS-{CDs-[TbL]} (*Synthesis_1*) upon seven heating-cooling cycles under alternating temperatures of 35 °C and 45 °C. (**c**) Temperature dependence of 450 nm (1) and 547 nm (2) bands of PSS-{CDs-[TbL]} (*Synthesis_2*) upon two heating-cooling cycles (first cycle colored by black and blue for 450 nm and 547 nm; second cycle colored by red and purple for 450 nm and 547 nm, respectively). (**d**) I_547_/I_450_ value versus temperature within the range of 25–50 Celsius degrees. (**e**) Emission spectra of the PSS-{CDs-[TbL]} (*Synthesis_2*) (l_ex_ = 360 nm) as a function of temperature over the 25–50 °C range. (**f**) Normalized fluorescence intensity upon four heating-cooling cycles of PSS-{CDs-[TbL]} (*Synthesis_2*) under alternating temperatures of 30 °C and 50 °C.

**Table 1 nanomaterials-11-03080-t001:** The average hydrodynamic diameters (d), electrokinetic potential values (ζ) and polydispersity indices (PDI) evaluated from the DLS measurements of PSS-{CDs-[TbL]} (*Synthesis_1*).

Name	d, nm	PDI	Zp, mV
PSS[TbL]	193.3 ± 1.2	0.171 ± 0.011	−26.2 ± 0.6
PSS-{CDs-[TbL]}	285.2 ± 1.2	0.258 ± 0.015	−45.7 ± 0.7
CDs	1648 ± 523.1	0.951 ± 0.043	−39.9 ± 3.7

## Data Availability

Not applicable.

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
