# Peer review of "Single Excited Dual Band Luminescent Hybrid Carbon Dots-Terbium Chelate Nanothermometer"

_nanomaterials, 2021, doi:10.3390/nano11113080_

Round 1
Reviewer 1 Report
In this work, [TbL]+ and CDs with different luminescence properties are incorporated into PSS, and their interaction and fluorescence response to temperature changes are studied in detail. More importantly, the resulting PSS-{CDs-[TbL]} can be used as the bicomponent ratiometric nanothermometers, overcoming the shortcomings of traditional single-parameter thermometers. This work is favorable for the temperature sensing applications related to inhomogeneous systems. Therefore, I suggest that this manuscript be published in Nanomaterials after minor revisions.
The three questions are as follows:
1. The temperatures marked in Figure 9b are 35 oC and 45 oC, while 25 oC and 50 oC are described in the caption. In addition, the temperatures marked in Figure 9f are 30 oC and 50 oC, while the data described in the caption and the manuscript are 25 oC and 50 oC. Why are there the inconsistencies? Please check these data.
2. The authors said, “the fluorescence of the CDs exhibits the insignificant changes within the temperature range.” However, in Figure 9c, for 450 nm, the ordinate at 25 oC changed from 1.0 to 1.15 after a heating-cooling cycle. Could such a change be negligible?
3. The fluorescence emission of CDs for 450 nm described in Figure S4 belongs to the triplet emission? However, the triplet state is generally considered to emit phosphorescence rather than fluorescence. Therefore, please check whether the description in Figure S4 is wrong.
Author Response
On behalf of the authors I want to express gratitude for evaluation and comments of the Reviewers and the Editor on our manuscript. We have now taken care of the queries raised by the Reviewers and the Editor to a fullest possible extent and prepared a thoroughly revised manuscript and supplementary material. The modified version of the files has been uploaded now.
Reviewer 1.
In this work, [TbL]+ and CDs with different luminescence properties are incorporated into PSS, and their interaction and fluorescence response to temperature changes are studied in detail. More importantly, the resulting PSS-{CDs-[TbL]} can be used as the bicomponent ratiometric nanothermometers, overcoming the shortcomings of traditional single-parameter thermometers. This work is favorable for the temperature sensing applications related to inhomogeneous systems. Therefore, I suggest that this manuscript be published in Nanomaterials after minor revisions.
The three questions are as follows:
- The temperatures marked in Figure 9b are 35 oC and 45 oC, while 25 oC and 50 oC are described in the caption. In addition, the temperatures marked in Figure 9f are 30 oC and 50 oC, while the data described in the caption and the manuscript are 25 oC and 50 oC. Why are there the inconsistencies? Please check these data.
The authors thank the reviewer for the comment. Erroneous figure caption has been revised accordingly.
- The authors said, “the fluorescence of the CDs exhibits the insignificant changes within the temperature range.” However, in Figure 9c, for 450 nm, the ordinate at 25 oC changed from 1.0 to 1.15 after a heating-cooling cycle. Could such a change be negligible?
Such changes are detectable under their referencing to the initial emission band of CDs, while the intensity ratio of the bands at 547 and 450 nm makes the aforesaid changes negligible under monitoring the thermally induced changes through I547/I450. To make this point clearer the corresponding sentence has been rewritten as the following: “…while the fluorescence of the CDs exhibits the insignificant changes within the temperature range in comparison with the intensity changes of the green band at 547 nm (Figure 9e).”
- The fluorescence emission of CDs for 450 nm described in Figure S4 belongs to the triplet emission? However, the triplet state is generally considered to emit phosphorescence rather than fluorescence. Therefore, please check whether the description in Figure S4 is wrong.
According to the literature data [Nano Research 2015, 8(2): 355–381 DOI 10.1007/s12274-014-0644-3] the actual mechanism of the photoluminescence (PL) of fluorescent carbon dots (CDs) is still an open debate among researchers. Moreover, wide emission peak of CDs derives from the diversity of PL centers, in turn, resulted from the inhomogeneous chemical structure of CDs. Thus, it is better to designate the level in Figure S4 as “the excited levels of CDs”.
Reviewer 2 Report
The manuscript by Zairov et al. reported the preparation of a single excited dual band luminescent nanothermometer through the combination of CDs and Tb3+ complex and then coating with polystyrenesulfonate. Though a variety of data were presented, the work suffers from several severe drawbacks.
Conceptual: I do not see the strong reason why the carbon dots and terbium chelate must be integrated into a single particle. The authors claimed that the coordinative binding between Tb3+ centers and the surface groups of CDs is the main driving force of their joint incorporation into the PSS-based nanobeads. However, no solid proofs were provided. If one only needs two photoluminescent components, then why carbon dots and terbium chelate. I think the authors should give deeper insight at this point to strengthen the necessity of carrying out this work.
Experiments: Both the purification of carbon dots and the PSS-based complexes/nanoparticles is too simple. Only filtration and centrifugation can not remove the small molecules involved in the system. This makes the physics and chemistry behind the material very vague. Figure 1 is more like a pure illustration, with the actual structure not proved.
Applications: For applications, at least the stability of the materials as a function of time should be evaluated. Is there fluorescence quenching (especially considering that small molecules are possibly present in the system due to the insufficient purification)? Besides, The working environment of the nanothermometer is the thermal sensing in biological samples. whether the Tb3+ complex is toxic or not, there is no relevant experimental proof.
Other issues:
- The characterization of the CDs, PSS-{CDs-[TbL]} colloids and interactions between CDs and [TbL]+ is insufficient, the main characterization methods are only TEM, PL and DLS.
- The quality of the figures is poor. They are too vague and do not provide enough annotations to explain what the picture expresses. In addition, some of the figures are hard to understand, such as Figure 3c and Figure 5c(why there are two lines?), Figure 9c (the lines are hard to be distinguished).
Author Response
On behalf of the authors I want to express gratitude for evaluation and comments of the Reviewers and the Editor on our manuscript. We have now taken care of the queries raised by the Reviewers and the Editor to a fullest possible extent and prepared a thoroughly revised manuscript and supplementary material. The modified version of the files has been uploaded now.
Conceptual: I do not see the strong reason why the carbon dots and terbium chelate must be integrated into a single particle. The authors claimed that the coordinative binding between Tb3+ centers and the surface groups of CDs is the main driving force of their joint incorporation into the PSS-based nanobeads. However, no solid proofs were provided. If one only needs two photoluminescent components, then why carbon dots and terbium chelate. I think the authors should give deeper insight at this point to strengthen the necessity of carrying out this work.
In accordance with the Reviewer’s criticism we have revised and reorganized the Introduction section in order to provide more sound justification of carrying out the work. It has been, in particular, highlighted that a progress in both sensitivity and selectivity of temperature sensing can be achieved on a basis of ratiometric sensing. Developing of ratiometric sensor can be realized through a combination of luminophoric blocks with different mechanisms responsible for generation of luminescent response on the same external stimuli. The different nature of lanthanide chelates and CDs, as well as a possibility to link these components through electrostatic and coordinative binding provides a prerequisite for their applicability as the components of the ratiometric sensor. The key role of the counter-ion binding on the self-aggregation of polyelectrolytes makes the self-aggregated polyelectrolytes convenient nanobeads open to incorporation of both cationic and anionic species for developing of ratiometric sensing systems.
Experiments: Both the purification of carbon dots and the PSS-based complexes/nanoparticles is too simple. Only filtration and centrifugation can not remove the small molecules involved in the system. This makes the physics and chemistry behind the material very vague. Figure 1 is more like a pure illustration, with the actual structure not proved.
The key property of the PSS-based hybrid system is worth noting as the answer on the issue raised by the Reviewer. Indeed, the as synthesized CDs colloids are admixed by small molecules, which can’t be washed out through a phase separation due to high hydrophilicity and small size of CDs, while the PSS-stabilized colloids exhibit easy phase separation through centrifugation. This facilitates washing out of the undesired small molecules, including residual CDs, TEA, unbound Tb(III) ions, from the nanoprecipitates consisting from water insoluble Tb(III) complexes and CDs bound with Tb(III) complexes incorporated into the PSS-based aggregates. The revised MS provides the additional sentence with this information (highlighted by red).
The Reviewer is right that Figure 1 provides the illustration of nanoarchitecture, which demonstrates two structural features of the colloids, which are core-shell nanostructure and binding between [TbL] and CDs within the PSS-shell. The demonstrated features derive from the well estimated following facts: (1) the well-known fact that high negative electrokinetic potential of polyelectrolyte-stabilized hybrid nanoparticles results from the self-assembly of polyelectrolyte molecules onto hard colloidal species; (2) carbon dots itself are too hydrophilic and negatively charged to be incorporated into PSS-[TbL] colloids, and their incorporation argues for the binding event.
Applications: For applications, at least the stability of the materials as a function of time should be evaluated. Is there fluorescence quenching (especially considering that small molecules are possibly present in the system due to the insufficient purification)? Besides, The working environment of the nanothermometer is the thermal sensing in biological samples. whether the Tb3+ complex is toxic or not, there is no relevant experimental proof.
The as synthesized hybrid PSS-stabilized colloids demonstrate invariant in time dual blue-green luminescence and reproducible luminescence response on temperature changes. As it was shown at figure 9b and 9f PSS-{CDs-[TbL]} system demonstrated excellent stability during 4 hours experiments upon six or seven heating-cooling cycles under alternating temperature of 25oC and 50oC. The invariance in luminescence intensity for up to 3 days of storage evidence steadiness of water colloids (Figure S5). The stability derives from the fundamental backgrounds of the polyelectrolyte-stabilized colloids, which are well documented in the work of Sukhishvili (ref. 13).
The low cytotoxicity and convenient cell internalization of the PSS-stabilized colloids based on the terbium complexes with calix[4]arene 1,3-diketones has been previously reported (ref. 10). This information has been included into the revised MS.
Other issues:
- The characterization of the CDs, PSS-{CDs-[TbL]} colloids and interactions between CDs and [TbL]+ is insufficient, the main characterization methods are only TEM, PL and DLS.
The dual luminescence of PSS-{CDs-[TbL]} colloids after the repeated phase separation is the key reason for the incorporation of both luminescent components into the PSS-nanobeads characterized by negative electrokinetic potential. The PL is the most convenient method to reveal the binding event between the terbium chelates and CDs. This is the prerequisite for the incorporation of CDs into the nanoprecipitated terbium chelate species, which are evident from the TEM images.
- The quality of the figures is poor. They are too vague and do not provide enough annotations to explain what the picture expresses. In addition, some of the figures are hard to understand, such as Figure 3c and Figure 5c(why there are two lines?), Figure 9c (the lines are hard to be distinguished).
The Figure Captions have been revised to explain that two sets of experimental points in Figs. 3c and 5c derive from heating and cooling.
Reviewer 3 Report
The author reported a joint incorporation of hybrid polyelectrolyte-stabilized colloids combining blue emitting citrate carbon dots and green emitting [TbL]+ chelate building blocks into the polysodium polystyrenesulfonate (PSS) aggregates. The synthetic conditions can be represented as a tool for tuning the fluorescent response upon heating and cooling within 25-50 oC. This can be used to developing either dual band luminescent colloids or transform the colloids into ratiometric temperature sensors via simple concentration variation. This work is interesting, it can be accepted after follow revision:
- page 7, line 10, the author state that the water solution of CDs has one distinct absorption bands. In fact, there are two bands at 350 nm and 270 nm. The author should point out the distinct one.
- Page 8, figure3C shows the Changes in the fluorescence intensity of the CDs as a function of temperature over the 25-70 oC range. However, this is not a perfect linear response for temperature. And the data points are not enough to support the linear fitting. The author should add more data point. The author should give a explanation at inflectionpoint (50 oC. figure3C )
- Page 8, figure3C shows the Changes in the fluorescence intensity of the CDs as a function of temperature over the 25-70 oC range. However, this is not a perfect linear response for temperature. And the data points are not enough to support the linear fitting. The author should add more data point. The author should give a explanation at inflectionpoint (50 oC, figure3C )
- Page 10, figure5C does not shows linear response for temperature, particularly for cooling process.
- The author should add time-resolved luminescence of the complex under the varied concentration of the CDs in the DMF solutions to well explain the corresponding mechanism.
Author Response
On behalf of the authors I want to express gratitude for evaluation and comments of the Reviewers and the Editor on our manuscript. We have now taken care of the queries raised by the Reviewers and the Editor to a fullest possible extent and prepared a thoroughly revised manuscript and supplementary material. The modified version of the files has been uploaded now.
The author reported a joint incorporation of hybrid polyelectrolyte-stabilized colloids combining blue emitting citrate carbon dots and green emitting [TbL]+ chelate building blocks into the polysodium polystyrenesulfonate (PSS) aggregates. The synthetic conditions can be represented as a tool for tuning the fluorescent response upon heating and cooling within 25-50 oC. This can be used to developing either dual band luminescent colloids or transform the colloids into ratiometric temperature sensors via simple concentration variation. This work is interesting, it can be accepted after follow revision:
- page 7, line 10, the author state that the water solution of CDs has one distinct absorption bands. In fact, there are two bands at 350 nm and 270 nm. The author should point out the distinct one.
Thank you, indeed, there are two absorption bands, which has been noted in the revised MS.
- Page 8, figure3C shows the Changes in the fluorescence intensity of the CDs as a function of temperature over the 25-70 oC range. However, this is not a perfect linear response for temperature. And the data points are not enough to support the linear fitting. The author should add more data point. The author should give a explanation at inflectionpoint (50 oC. figure3C )
Additional experiment was performed to retrieve required data points. Figure 5c was refined by additional points with temperature step of 5 Celsius degrees. With these clarifications authors claim linearity of Ilum in the range of 25-70° C.
- Page 10, figure5C does not shows linear response for temperature, particularly for cooling process.
Yes, Figure 5c indicate real thermal responsivity in wide temperature range (25-70° C). The heating up to 70° C can provide the temperature-induced dissolution of the nanoprecipitated terbium chelates. However, the deviation is insignificant, which is the fact indicating the stability of PSS-[TbL], while the linearity and repeatability is much greater in the lower physiological temperature range (35-45° C).
- The author should add time-resolved luminescence of the complex under the varied concentration of the CDs in the DMF solutions to well explain the corresponding mechanism.
The authors have added these experimental data (Table S1), which show the small but detectable increase in the lifetime of the terbium-centered luminescence values under the increased concentrations of the CDs at the constant concentration of the terbium chelates in the DMF solutions. This argues for the binding between the terbium chelates and CDs as the prerequisite for their joint nanoprecipitation.
Reviewer 4 Report
Editor
Nanomaterials
The manuscript (Ref: nanomaterials-1452383) of the paper entitled " Single excited dual band luminescent hybrid carbon dots-terbium chelate nanothermometer", in my opinion can be considered for publication in the journal “Nanomaterials” after its minor revision. The article shows very interesting experimental results concerning investigation of spectroscopic properties of novel and complex nanomaterials, combining carbon quantum dots and lanthanide complexes for high-sensitivity (up to 7%/K) luminescence temperature sensing. The following points should be addressed before publication of the article:
- The Authors should change the relative sensitivity from negative to positive values (if the measured spectroscopic parameter decreases with temperature, the derivative results or the final sensitivity values should be multiplied by "-1"), in order to provide the physical meaning for the data obtained.
- I suggest to update the literature part, by revising and adding some recent papers dealing with luminescence temperature sensing with non-thermally coupled levels of lanthanides, as well as some comparisons and the relevant discussion
Author Response
On behalf of the authors I want to express gratitude for evaluation and comments of the Reviewers and the Editor on our manuscript. We have now taken care of the queries raised by the Reviewers and the Editor to a fullest possible extent and prepared a thoroughly revised manuscript and supplementary material. The modified version of the files has been uploaded now.
The manuscript (Ref: nanomaterials-1452383) of the paper entitled " Single excited dual band luminescent hybrid carbon dots-terbium chelate nanothermometer", in my opinion can be considered for publication in the journal “Nanomaterials” after its minor revision. The article shows very interesting experimental results concerning investigation of spectroscopic properties of novel and complex nanomaterials, combining carbon quantum dots and lanthanide complexes for high-sensitivity (up to 7%/K) luminescence temperature sensing. The following points should be addressed before publication of the article:
- The Authors should change the relative sensitivity from negative to positive values (if the measured spectroscopic parameter decreases with temperature, the derivative results or the final sensitivity values should be multiplied by "-1"), in order to provide the physical meaning for the data obtained.
The relative sensitivities have been changed to positive by multiplying by -1. The figure has been redrawn accordingly.
- I suggest to update the literature part, by revising and adding some recent papers dealing with luminescence temperature sensing with non-thermally coupled levels of lanthanides, as well as some comparisons and the relevant discussion
The authors thank the Editor for valuable comment. Indeed, lanthanide complexes with organic ligands of different structure, including those incorporated in the composition of nanoparticles, represent promising basis for thermo-responsive nanomaterial because of the presence of different mechanisms, with respect to those regulating the thermo-responsive behavior of lanthanide ions in the inorganic matrices Following paragraph was added in the very beginning of the manuscript.
“Wide diversity of organic and inorganic luminophores allows to apply different mechanisms to generate luminescent response on external stimuli, which results in the great diversity of sensing systems [[10.1246/cl.2004.1438, 10.1039/D1TC00709B, 10.1016/j.jallcom.2021.158891]. Progress in sensitivity and selectivity of sensing is real challenge, which requires further evolution of sensing systems. Ratiometric approach based on generation of two emission bands with different sensitivity on specific external stimuli opens great opportunities in evolution of sensing systems [10.1021/nn405456e; 10.1039/C9TC02328C].”